



# Relationship between low-cloud presence and the amount of overlying aerosols

Chul E. Chung[1], Anna Lewinschal[2] and Eric Wilcox[1]

[1]Desert Research Institute, Reno, 89512, USA
[2]Department of Meteorology and Bolin Centre for Climate Research, Stockholm University, 106 91 Stockholm, Sweden

*Correspondence to*: C. E. Chung (Eddy.Chung@dri.edu)

**Abstract.** Aerosols are often advected above cloud decks, and the amount of aerosols over cloud has been assumed to be similar to those at the same heights in nearby clear sky. In this assumption, cloud and aerosol above cloud top height are considered randomly located with respect to each other. The Cloud-Aerosol Lidar and Infrared Pathfinder Satellite Observations (CALIPSO) data are analyzed here to investigate this assumption on global scales.

The CALIPSO data reveal that the aerosol optical depth (AOD) above low cloud tends to be smaller than in nearby clear sky during the daytime and the opposite is true during the nighttime. In particular, over oceanic regions with wide-spread low cloud, such as the tropical southeastern Atlantic Ocean and northeastern Pacific Ocean, the daytime AOD above low cloud is often 40% smaller than in surrounding clear skies.

## 1. Introduction

The magnitude of the global aerosol direct radiative forcing (due to absorption and scattering of incoming sunlight) has been estimated to range from -0.85 to 0.15 W m$^{-2}$ in the 5$^{th}$ Intergovernmental Panel on Climate Change report (Myhre et al., 2013a). This uncertainty range is based partly on aerosol simulations (Myhre et al., 2013b) and also on semi-empirical studies (e.g. Myhre, 2009; Su et al., 2013) where aerosol observations were used to constrain the aerosol simulations.

A part of this uncertainty range stems from uncertainties related to the vertical distribution of aerosols. Radiation modeling studies (Haywood and Shine, 1997; Meloni et al., 2005; Choi and Chung, 2014) showed that the sensitivity of aerosol forcing to the aerosol vertical profile arises mainly as a consequence of the location of absorbing particles relative to clouds. Absorbing aerosols above clouds absorb both downwelling solar radiation and upwelling radiation reflected from the underlying clouds, giving stronger absorption and more positive forcing than in the absence of clouds or if the aerosol is located below cloud. Thus, it is important to determine as accurately as possible the amount of absorbing aerosols above clouds. This also means that cloud amount and albedo become important factors in determining the forcing of absorbing aerosols. Chand et al. (2009) found that for biomass burning smoke (with a single scattering albedo, SSA, of 0.85) over the southeastern Atlantic stratocumulus clouds, the net radiative forcing changes the sign from negative to positive when the cloud fraction exceeds 40%.

When global aerosol simulations were constrained by clear-sky AOD derived from MODderate Resolution Imaging Spectroradiometer (MODIS) to reduce the uncertainty in aerosol radiative forcing estimates (e.g. Chung et al., 2005; Myhre, 2009; Su et al., 2013), the AOD was assumed to be homogeneously distributed



within a grid cell. Clouds, on the other hand, do not necessarily occupy the entire grid cell, and are prescribed over a portion of the grid cell. This procedure thereby assumes that the AOD is independent of clouds and that the AOD above clouds is the same as that at the same height in the clear sky neighborhoods. The main goal of the present study is to validate this assumption with observations. As mentioned above, the radiative forcing of

absorbing aerosols is enhanced above bright surfaces such as clouds. Thus, if absorbing aerosols are actually less or more abundant above clouds compared to the clear sky, prescribing a horizontally uniform clear-sky AOD over the entire grid cell would lead to an over- or under-estimation of the aerosol direct radiative forcing.

Looking into the amount of absorbing aerosol above cloud has additional scientific significance. Absorbing aerosols above clouds can introduce biases in satellite retrieved cloud effective radius and optical

depth (Haywood et al., 2004). In addition to improving the estimates of aerosol direct radiative forcing, accurate estimates of AOD above clouds and any systematic difference between the AOD above clouds and that in surrounding clear skies could help to improve cloud retrievals from passive satellite sensors.

If aerosols are actually less or more abundant above clouds compared to the clear sky neighborhoods, what could the possible reasons be? One candidate is large-scale circulation patterns. For instance, advection or

emissions of aerosols could be caused by specific circulation patterns or transport of air masses that would also either promote or inhibit low cloud formation (e.g., Mauger and Norris, 2007). Another candidate is aerosol semi-direct effects. Previous studies that have focused on marine stratocumulus cloud regions have shown that absorbing aerosols above the boundary layer can increase the low-cloud liquid water path (e.g., Johnson et al., 2004 who used a large eddy model to simulate conditions off the Californian coast, i.e., stratocumulus clouds

with tops between around 700 m; Wilcox 2010 who studied the southeastern Atlantic stratocumulus clouds with tops typically below 1.3 km) and low-cloud fraction over the south east Atlantic region (Costantino and Bréon, 2013 who defined low clouds as clouds with the tops below 600 hPa). While the suggested connection between aerosols and cloud liquid water path or cloud fraction might not be applicable on a global scale, these studies showed that aerosols can affect cloud properties through semi-direct effects in a manner that might manifest as a

difference between AOD above clouds and that in the clear sky neighborhoods in certain regions.

The Cloud-Aerosol Lidar and Infrared Pathfinder Satellite Observations (CALIPSO) offer global aerosol and cloud vertical profile observations (Winker et al., 2009). Moreover, the Cloud-Aerosol Lidar with Orthogonal Polarization (CALIOP) instrument onboard CALIPSO retrieves aerosol backscatter from space, both in clear and cloudy skies, and thereby offers an ideal platform for determining aerosol amount above clouds.

Devasthale et al. (2011) used CALIPSO data to investigate the occurrence of aerosols above liquid water clouds. In an inter-comparison of above-cloud AOD from A-train satellites sensors, Jethva et al. (2014) found that the CALIPSO derived above-cloud AOD was consistently lower than those derived from the other satellites sensors in the A-Train. Chand et al. (2008) and Liu et al. (2015) both used alternative retrieval methods to the standard CALIPSO algorithm to derive AOD above clouds specifically. Other efforts to quantify above-cloud AOD from

passive satellite sensors include the works by e.g., Knobelspisse et al. (2011), Meyer et al. (2013), Torres et al. (2012) and Waquet et al. (2009, 2013).

Here, we use CALIPSO data and compare the observed AOD above low clouds with that at the same height in the nearby clear sky, to validate the aforementioned assumption that aerosols and clouds are randomly distributed. If aerosols and clouds are randomly located, the above-cloud AOD would be equal to that in clear

sky at the same height. In other words, if the aerosol distribution is independent of the presence of clouds, the





AOD at any altitude should be the same whether it is measured in clear sky or cloudy sky. If the aerosol distribution co-varies with that of clouds, either through the semi-direct effects or through large-scale atmospheric circulation, systematic differences between the AOD above clouds and that in clear sky should be visible. We utilize CALIPSO observations to investigate the AOD above cloud-top-height in relation to the presence of clouds on a global scale. The focus is solely on AOD above clouds-top-height, and thus aerosol indirect effects on cloud microphysics are not addressed in the present study.

The layout of the study is as follows. The CALIPSO data and retrieval algorithms relevant for the present work are described in Section 2. Section 3 describes the method used to derive the global above cloud AOD from CALIPSO standard data products. The results are presented in Section 4, followed by a discussion and conclusions in Section 5.

## 2. Data

The CALIPSO satellite was launched in April 2006 as a part of the A-train satellite constellation and has collected atmospheric data since June 2006. The satellite has a 16 days repeat cycle and collects global data between 82°N and 82°S. The CALIOP instrument on board the CALIPSO satellite is a near- nadir viewing lidar that collects vertical profiles of atmospheric backscatter data. The lidar instrument transmits polarized light at the wavelengths 532 nm and 1064 nm and measures the range resolved backscatter intensity for these wavelengths, including the amount of depolarization (i.e., backscatter signal polarized perpendicular to the outgoing beam) at 532 nm. CALIPSO Level 1 data consists of calibrated attenuated backscatter data at a resolution of 1/3 km in the horizontal direction along the ground track and 30 m in the vertical below 8.2 km. Atmospheric features, such as clouds and aerosols, can be identified in the backscatter data based on their enhanced backscatter intensity compared to the background molecular backscatter from a clear atmosphere. An overview of the CALIPSO mission can be found in the work by Winker et al. (2009).

CALIPSO level 2 data is derived with a collection of algorithms that identify and classify atmospheric feature layers and calculate their atmospheric extinction. Detailed descriptions of the CALIPSO data acquisition and processing are provided by Hu et al. (2009), Liu et al. (2009), Omar et al. (2009), Vaughan et al. (2009), and Young and Vaughan (2009). The layer detection algorithm uses an iterative, multi scale approach when searching for cloud and aerosol layers in the backscatter data (Vaughan et al., 2009). The algorithm is designed to retain as much as possible of the horizontal inhomogeneities in the backscatter data, while performing a minimum of horizontal averaging to increase the Signal to Noise Ratio (SNR), so that the backscatter of a feature, if present, exceeds the detection threshold. The layer detection algorithm identifies the height of the top and the bottom of the feature layers, i.e. the bounding altitudes of the area where the backscatter signal exceeds that of the background atmosphere. Thus, an altitude dependent threshold value array for enhanced backscatter has to be determined for each lidar profile. This threshold value array varies with background illumination, which depends both on solar radiation and surface albedo (Winker et al., 2009). During daytime, the background noise is large which makes detection of tenuous aerosol layers more difficult compared to detection in nighttime retrievals. During daytime, faint aerosol layers can remain undetected, or their geometrical thickness can be underestimated to a larger extent than during nighttime. This may, in turn, lead to an underestimation of the total daytime AOD compared to that derived from nighttime data (Rogers et al. 2014). Moreover, during daytime,





reflection of solar radiation from clouds can contribute to the background noise in the lidar profiles, and could, in a similar manner, lead to undetected faint aerosol layers above clouds (Kacenelenbogen et al., 2014).

The layers that are identified by the iterative layer detection algorithm are separated into cloud and aerosol layers based on their backscatter signals in all three channels, as well as their altitude and geographical position (Liu et al., 2009). However, the cloud-aerosol separation algorithm is applied for features detected at a horizontal averaging resolution of 5km or lower. All features detected at one single shot are classified as clouds. In regions with large dust emissions this can lead to misclassifications of dense dust layers as clouds.

Features identified as aerosols are further categorized according to aerosol type (Omar et al., 2009). The CALIPSO algorithm uses six different characteristic aerosol models to represent atmospheric aerosols: dust, polluted dust, smoke, clean marine, clean continental, and polluted continental. These representative aerosol models include assumptions about the particle size distribution as well as the optical properties of the aerosol. The aerosol extinction coefficients are derived based on the aerosol model and the corresponding representative lidar ratio associated with each aerosol type. Thus, the total aerosol extinction and AOD is dependent on the accuracy of the aerosol model and that the correct aerosol model is chosen for an aerosol layer.

## 3. Methodology

To investigate aerosol abundance above low clouds, we use the CALIPSO level 2 data to calculate the AOD above the maximum low-cloud top height in each grid cell in clear sky as well as the AOD above low clouds on a global $2° \times 5°$ latitude-longitude grid. We use version 3.01 of the level 2 data which covers the time period from June 2006 to October 2011. The daytime and nighttime data are analyzed separately. We focus primarily on daytime data here because aerosol direct radiative forcing is due almost entirely to the absorption and scattering of solar radiation. However, nighttime data which have higher SNR, is included for reference. We study the AOD above low clouds since aerosols are typically advected above low clouds, and the low-cloud-top height provides a more uniform reference altitude than a combination of all clouds.

In this study, the aerosol profile product and cloud layer products are used. The horizontal resolution of the level 2 aerosol profile product is 5 km along the satellite ground track, while cloud layer products are available at 5 km and 1/3 km (corresponding to one single lidar shot) horizontal resolutions. The cloud layer products are utilized to identify and separate 5 km horizontal resolution clear sky aerosol extinction profiles from profiles containing low clouds. In the CALIPSO data products, low clouds are defined as clouds with tops below 680 hPa (Liu et al., 2005). The detection ability of the CALIPSO feature detection algorithm is dependent on the spatial averaging of the lidar backscatter data. Tenuous features typically require averaging over a large horizontal distance to be detected, while more opaque layers require less horizontal averaging to be detected. The 5 km cloud layer product contains all the cloud layers detected by the CALIPSO feature detection algorithm at horizontal averaging resolutions of 5 km or lower in the iterative feature scanning procedure (cf. Vaughan et al. 2009 for a detailed description of how atmospheric features are identified). Cloud layers detected at higher horizontal resolution (at one single lidar shot) are reported separately in the 1/3 km cloud layer product. Clouds detected in one single shot are typically optically thick boundary layer clouds, e.g. stratocumulus (Vaughan et al., 2009). However, in desert regions dense dust layers are sometimes detected in one single shot but are reported in the 1/3 km cloud layer product. Thus, part of the low clouds identified in these regions might be dust aerosols.



We use a combination of the 5 km and 1/3 km cloud layer products to obtain the most accurate low cloud distribution. A 5 km resolution low-cloud fraction is calculated from the 1/3 km cloud layer product (based on 15 consecutive single lidar shots) if the 5 km cloud layer product reports a cloud free 5 km profile. Both the 5 km and the 1/3 km cloud layer products contain information about the cloud-top height for every cloud layer, necessary for calculating the above cloud-top height AOD variables in this study. Only profiles free from middle and high clouds (i.e. clouds with tops above 680hPa) are included in the analysis.

The aerosol extinction coefficient at 532 nm from the two-dimensional aerosol profile product is aggregated onto a three-dimensional grid with a horizontal latitude-longitude resolution of 2°×5°. The 5 km horizontal resolution aerosol extinction profiles are separated into clear sky profiles and profiles containing low clouds. The classification procedure is illustrated in Fig. 1. The aerosol extinction profiles for which the 5 km cloud layer product reports a cloud layer are categorized as overcast (column 2 in Fig. 1). However, profiles where the 5 km cloud layer product reports a cloud free column can still contain cloud layers detected at higher resolution (i.e. clouds reported in the 1/3 km cloud layer product). If the 5 km cloud layer product reports a cloud free 5 km profile, the 5 km resolution cloud fraction derived from the 1/3 km cloud layer product is checked. If this cloud fraction is one or zero, the profile is categorized as overcast (column 4 in Fig. 1) or cloud free (column 3 in Fig. 1), respectively. If the 5 km cloud fraction is less than one but larger than zero, the aerosol extinction is weighted by the cloud fraction into clear and cloudy profiles accordingly (column 1 in Fig. 1). Two types of grid-cell average monthly mean aerosol extinction coefficient profiles, one cloudy and one clear-sky, are then calculated for each 2°×5° grid cell, based on all the cloudy and clear 5 km aerosol extinction coefficient profiles collected in each 2°×5° grid cell during one month.

The two types of monthly mean aerosol extinction coefficient profiles are integrated through the atmosphere down to the maximum low-cloud-top height found in each 2°×5° grid cell for each month and year. The maximum low-cloud-top height is defined as the single largest value of the low-cloud-top heights found in a 2°×5° grid-cell, considering both the 1/3 km and 5 km resolution cloud data. The maximum low-cloud-top height is chosen as reference height to eliminate spurious effect arising from aerosol sampling deficiencies in cloudy profiles at altitudes below the maximum cloud-top height. Thus, a grid-cell average AOD above low clouds and an AOD in clear sky are derived above the same maximum low-cloud-top height detected in each 2°×5° grid cell. The AOD integrated down to the maximum low-cloud-top height, both in clear sky and when low clouds are detected, will hereafter be referred to as $AODct$. $\Delta AODct$ here will designate the difference between $AODct$ above low clouds and $AODct$ in clear sky, and positive values mean that $AODct$ above low clouds is larger than $AODct$ in clear sky.

In addition to the AOD, we calculate the low-cloud fraction for each 2°×5° grid cell. This two-dimensional low-cloud fraction is derived from the fraction of 5 km resolution profiles that contain low clouds combined with the 5 km resolution low-cloud fraction derived from the 1/3 km cloud layer product described above.

We apply a data quality control similar to that of Winker et al. (2013). Only aerosol or cloud samples with a Cloud Aerosol Discrimination (CAD) score with an absolute value equal to or greater than 20 are used in the analysis. Moreover, we screen the data for the Extinction Quality Control (QC) flag. Only samples with QC values corresponding to 0, 1, 16 and 18 are included in the analysis (cf. Winker et al., 2013). Screening for



extinction uncertainty is done to ensure that profiles containing samples with an uncertainty of 99.9 km$^{-1}$, which indicates failed retrieval with respect to the uncertainty estimate, are not included.

## 4. Results

Figs. 2 and 6 display the multi-year seasonal average difference between AOD above the maximum low-cloud-top height when and where low clouds are present and AOD above the same height in clear sky, denoted as $\Delta AODct$. A large number of cloudy and clear air profiles were used to derive the $AODct$. Fig. 3 shows how many profiles were used over each gridbox. The maximum low-cloud-top height from daytime retrievals is shown in Fig. 4. Only the gridboxes where the numbers of cloudy and clear profiles are both larger than 5% of the total number of profiles in the gridbox are shown in Figs. 2 and 6. Thus, many of the potentially spurious cloud samples over desert regions (cf. Section 2), where the low cloud fraction (shown in Fig. 5) is small, are removed. We also exclude the gridboxes in Figs. 2 and 6 that have $\Delta AODct < 1.00 \times 10^{-2}$.

The daytime difference between $AODct$ above low clouds and $AODct$ in clear sky, as shown in Fig. 2, displays both large-scale spatial patterns and seasonal variations. The difference is most pronounced in the tropical regions, where the $AODct$ above low clouds is largest. The nighttime $\Delta AODct$ (Fig.6) is generally smaller and noisier than the daytime $\Delta AODc$ field. Similarly to the daytime global $\Delta AODct$ distribution, the nighttime $AODct$ difference between cloudy and clear skies is mainly concentrated to the tropical regions, except during March, April and May (MAM) and June, July and August (JJA), when pronounced $AODct$ differences between cloudy and clear sky are found also in the northern hemisphere extratropics. The nighttime $AODct$ above low clouds also shows higher values in the northern hemisphere extratropical regions compared to that derived from daytime retrievals.

Are $\Delta AODct$ values in Figs. 2 and 6 statistically significant? Applying Student's $t$-test to each gridbox, as commonly done in the literature, we find that they are not significant. However, we need to note that applying Student's $t$-test to each grid assumes that grids are independent of each other (Decremer et al., 2014), which is not correct. When grids are related, the bar for significance should be lower than when grids are independent, meaning that applying Student's $t$-test to each grid could determine actually significant values to be statistically insignificant. Some of $\Delta AODct$ values in Figs. 2 and 6 have large-scale spatial patterns, which means that grids are related to each other. Attempts have been made to address spatial correlation in the statistics community (Elmore et al., 2006), but such techniques cannot be used to determine the significance of local features. Thus, we first test the significance of the global average of daytime $\Delta AODct$ since the global average does not suffer the aforementioned grid-dependence issue. We find that the global average of $\Delta AODct$ is largest in September and also statistically significant using a one-sided 95% confidence interval in this calendar month, while statistically insignificant in the other months. If we limit the average to the 20°W-20°E & 30°S-Eq. area, where daytime $\Delta AODct$ is very large, the average of daytime $\Delta AODct$ is statistically significant in September and also November. Please note that an area average other a global average can still be related to the average of another region (though the relation would probably be less than using each grid) and so the standard significance testing is likely to underestimate the significance in this case. Overall, we conclude that $\Delta AODct$ contains statistically significant features.

The global average $AODct$ above low clouds is much larger during the nighttime than during the daytime for all seasons (Table 1), which is most likely related to a higher SNR for nighttime retrievals compared





to those during the daytime. The seasonal global mean values range from 0.011 to 0.029 for nighttime *AODct* above low clouds and between 0.007 and 0.020 for that from daytime retrievals. Both the nighttime and daytime *AODct* above low clouds are largest in JJA. The global average *ΔAODct* (calculated for gridboxes where the cloud fraction and clear air fraction exceeds 5%) from nighttime retrievals are positive for all seasons while the

opposite is true for that from daytime retrievals (Table 2). This finding is intact even if all the gridboxes are included in the calculation. As Table 2 shows, the daytime global average is largest during SON, at $-6.01\times10^{-3}$, and that for nighttime is largest during MAM, at $2.22\times10^{-3}$.

We further analyze the results in Figs. 2 and 6 by region and aerosol type. The aerosol types identified by the CALIPSO aerosol extinction retrieval algorithm (cf. Section 2) that contribute significantly to the global

daytime *AODct* above low clouds are dust, smoke and polluted dust (Table 1). The clean and polluted continental and clean marine aerosols, i.e., the other types, contribute negligibly to the daytime *AODct*. Dust is the aerosol type that contributes with the largest fraction to the daytime *AODct* above low clouds, with up to 65% and 70% in MAM and JJA, respectively. Dust also dominates the daytime *AODct* difference between clear sky and above low clouds, particularly in the abovementioned seasons. Over most parts of North Africa and the Arabian

Peninsula, the *AODct* above low clouds is substantially higher than that in clear sky. However, the dust *AODct* is mainly found over desert regions where the low cloud fraction is less than 5%, and thus, Δ*AODct* over most desert regions is not included in Figs. 2 and 6. *AODct* above low clouds is typically lower than *AODct* in clear sky where dust is transported over the Atlantic Ocean, except over the central tropical Atlantic Ocean in JJA (Fig. 7c). Dust aerosol transported over the Arabian Sea, on the other hand, has generally higher *AODct* above

low clouds than in clear sky. These daytime results for dust Δ*AODct* are mostly consistent with the nighttime Δ*AODc*t results, although the nighttime *AODct* has not been separated for different aerosol types.

In the boreal spring, dust and pollution are transported from the Eurasian continent over to the North Pacific Ocean (e.g. VanCuren, 2003; Husar et al., 2001; Merrill et al., 1989). Over the northeastern Pacific Ocean, the daytime *AODct* above low clouds due to polluted and pure dust is lower than the clear sky *AODct* for

these aerosol types (Figs. 7a and b). This difference can be observed both in MAM and JJA. A similar *AODct* difference for all aerosols between cloudy and clear skies is also visible in the nighttime data (Figs. 6b and c). Pollution over India contains aerosols from fossil fuel combustion and biomass burning (Gustafsson et al., 2009). The main contributions to the daytime *AODct* above low clouds in this region come from aerosols identified as polluted and pure dust in the CALIPSO data. The daytime dust *AODct* above low clouds is lower than *AODct* in

clear sky over India in JJA (Fig. 7c). Over Eastern China, where fossil fuel combustion is the dominating aerosol source (Chen et al., 2013), the *AODct* consists of contributions from smoke, dust and polluted dust. However, the daytime Δ*AODct* field over China is generally noisy. Similarly, the nighttime Δ*AODct* field over South and East Asia shows large spatial variations.

During the biomass burning season in southern Africa (August to October) smoke aerosol is advected

over the tropical southeastern Atlantic Ocean (e.g., Anderson et al., 1996; Edwards et al., 2006). Large amounts of low cloud in combination with the absorbing properties of smoke from biomass burning make this region one of the particular interests for quantifying AOD above clouds. In September, October and November (SON), daytime smoke *AODct* above low clouds derived from the CALIPSO data is lower than that in clear sky in this region (Fig. 7d). This is also the season when the magnitude of global average daytime Δ*AODct* is largest, (Table



2) and the contribution from smoke aerosol to the total daytime *AODct* above low clouds is largest, 23% (Table 1). However, a corresponding difference is not visible in the data from nighttime retrievals (Fig. 6d).

Similar to the daytime smoke *AODct* difference over the tropical Atlantic Ocean, the daytime smoke *AODct* in clear sky is larger than above low cloud over the South American continent, another region where biomass burning aerosols are emitted during SON. The corresponding nighttime Δ*AODct* in this region is noisier than the daytime Δ*AODct*.

### 5. Discussion and Conclusions

Like any observation, the CALIPSO data are subject to uncertainties. That said, we have offered the first observational quantification of the difference between AODs above cloud-top height in clear sky and where low clouds are detected.

The spatial distribution of the daytime difference between *AODct* in clear sky and above low clouds is found not to be random. Over ocean regions where low cloud fraction is large (i.e. the tropical southeast Atlantic Ocean and northeast Pacific Ocean), the daytime *AODct* in clear sky typically exceeds that above low clouds. These are also the areas where aerosols are frequently observed above clouds (Devasthale and Thomas, 2011).

In general, daytime *AODct* above low clouds is smaller than that in clear sky. The largest global average seasonal daytime difference is $-6.01 \times 10^{-3}$ during SON. On the other hand, the nighttime seasonal global average *AODct* above cloud is larger than that in clear skies, and the largest global average seasonal difference is $2.22 \times 10^{-3}$ in MAM. However, the nighttime Δ*AODct* spatial distribution is noisier than that derived from daytime data, and the magnitude of the nighttime global average Δ*AODct* is smaller than the corresponding daytime values.

The results presented in this study indicate that daytime AOD above low clouds is generally lower than that in clear sky at the same height over e.g., the southeastern Atlantic Ocean during SON. The local *AODct* above low clouds is up to 0.04 lower than that in clear sky, although the results of Wilcox (2010) and Costantino and Bréon (2013) suggest that absorbing aerosols above the boundary layer can increase cloud thickness and cloud fraction in this region. A possibility remains that other processes govern the difference between *AODct* in cloudy and clear sky, by either involving cloud-aerosol interactions or large-scale circulation patterns. However, a corresponding difference cannot be seen in the Δ*AODct* derived from nighttime retrievals.

Before any conclusion can be drawn concerning the physical relevance of the results presented here, further studies are necessary to determine the robustness of the results. The detection threshold in the feature detection algorithm varies depending on the background lighting conditions, and thus, on the presence of clouds but also on the surface albedo during the daytime (cf. Hunt et al. 2009; Vaughan et al. 2009; Chepfer et al. 2013; Kacenelenbogen et al. 2014). This means that the daytime *AODct* above low bright clouds reported here might be underestimated compared to the *AODct* in clear sky in the same grid cell. Thus, the negative daytime Δ*AODct* might simply be a result of systematic differences between the detection thresholds in clear sky and above low bright clouds. Moreover, it is possible that the lower albedo of the ocean surface compared to that of land enhances the *AODct* bias between clear and cloudy sky over the ocean. The absence of a pronounced difference between the *AODct* above low cloud and the *AODct* in clear sky over the southeastern Atlantic Ocean during SON derived from nighttime retrievals is consistent with this conjecture. However, differences in meteorological conditions as well as differences in the aerosol distribution during the daytime and nighttime might also





influence this $\Delta AODct$ day and night discrepancy. Over the northwest Pacific Ocean in MAM and western tropical Atlantic Ocean in JJA, on the other hand, the nighttime results indicate, in agreement with the daytime results, that $AODct$ above low clouds is lower than that in clear sky.

A solution to this problem could be to apply a similar approach as that of Chepfer et al. (2013), who

5 used a fixed detection threshold, which exceeds the variable detection threshold in the standard data processing algorithm, to derive an alternative cloud climatology from the CALIPSO backscatter data. The method requires a reprocessing of the level 1 backscatter data to obtain a new level 2 aerosol extinction data set. Deriving this alternative aerosol extinction level 2 data is not trivial, but probably necessary, for an unbiased comparison of daytime $AODct$ above low clouds and in clear sky.

10 **Acknowledgments**

This study was funded by the National Science Foundation (AGS-1455759). Data were obtained from NASA Langley Research Center Atmospheric Science Data Center. We thank the ICARE Data and Services Center for providing access to data used in this study.



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



**Tables**

| | Night | Day | | | |
|---|---|---|---|---|---|
| | *AODct* | *AODct* | Dust | Smoke | Polluted dust |
| DJF | 0.011 | 0.007 | 34% | 17% | 43% |
| MAM | 0.021 | 0.015 | 65% | 8% | 24% |
| JJA | 0.029 | 0.020 | 70% | 12% | 16% |
| SON | 0.019 | 0.010 | 47% | 23% | 27% |

5    **Table 1.** Global and seasonal averages of *AODct* above low clouds for nighttime and daytime and the fraction of the daytime data that is classified as dust, smoke or polluted dust.



| | Night | Day |
|---|---|---|
| | $\Delta AODct \times 10^{-3}$ | $\Delta AODct \times 10^{-3}$ |
| DJF | 1.91 (0.24) | -2.06 (-0.00) |
| MAM | 2.22 (0.22) | -3.00 (-0.44) |
| JJA | 1.60 (0.24) | -3.64 (-0.52) |
| SON | 1.06 (0.10) | -6.01 (-0.61) |

**Table 2.** Global and seasonal averages of $\Delta AODct$ for nighttime and daytime. The average is made over the gridcells where the $\Delta AODct$ is $> 1.00 \times 10^{-2}$, and the low cloud fraction and clear-sky fraction are both $> 5\%$. The numbers in parentheses are the averages over all gridcells where the cloud fraction and clear air fraction are both $> 5\%$, irrespective of the magnitude of $\Delta AODct$.




**Figures**

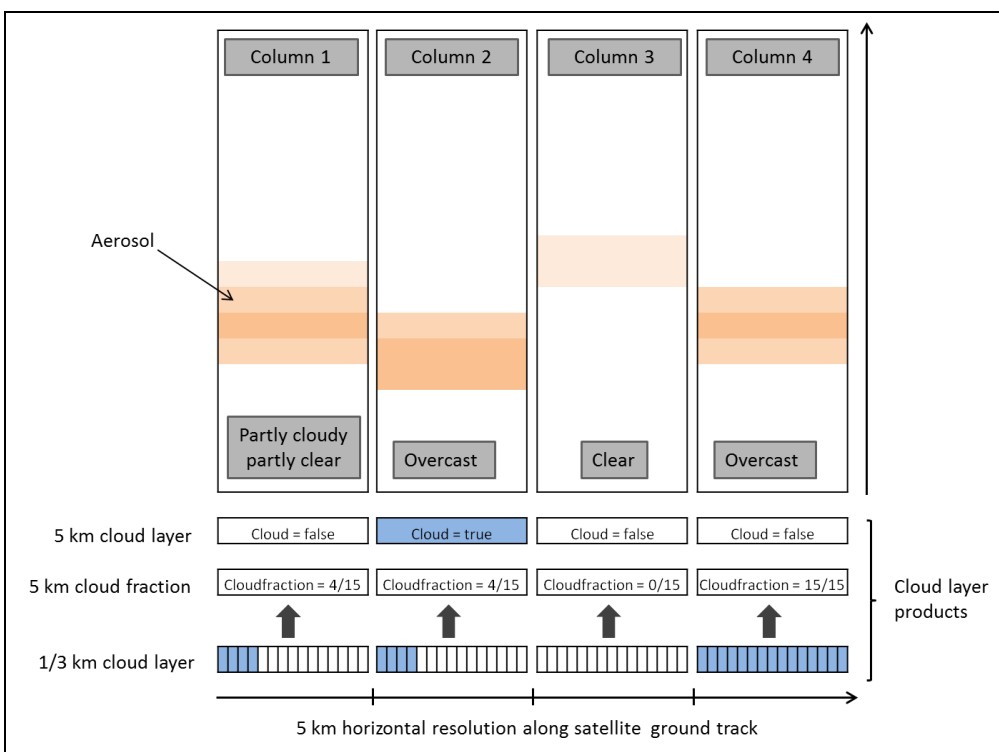

**Figure 1.** Illustration of the classification procedure of cloudy and clear sky aerosol extinction profiles. Only profiles where both the 1/3 and 5 km cloud layer products indicate clear sky are classified as completely clear (column 3). Only profiles where either the 5 km cloud layer product indicates a cloud or the 1/3 km layer data indicates a 5 km cloud fraction equal to 1 are considered completely cloudy or overcast (column 2 and 4 respectively). All profiles where the 5 km cloud layer product does not report a cloud, but the 5 km cloud fraction is larger than zero but less than 1, are considered partly cloudy and partly clear (column 1).





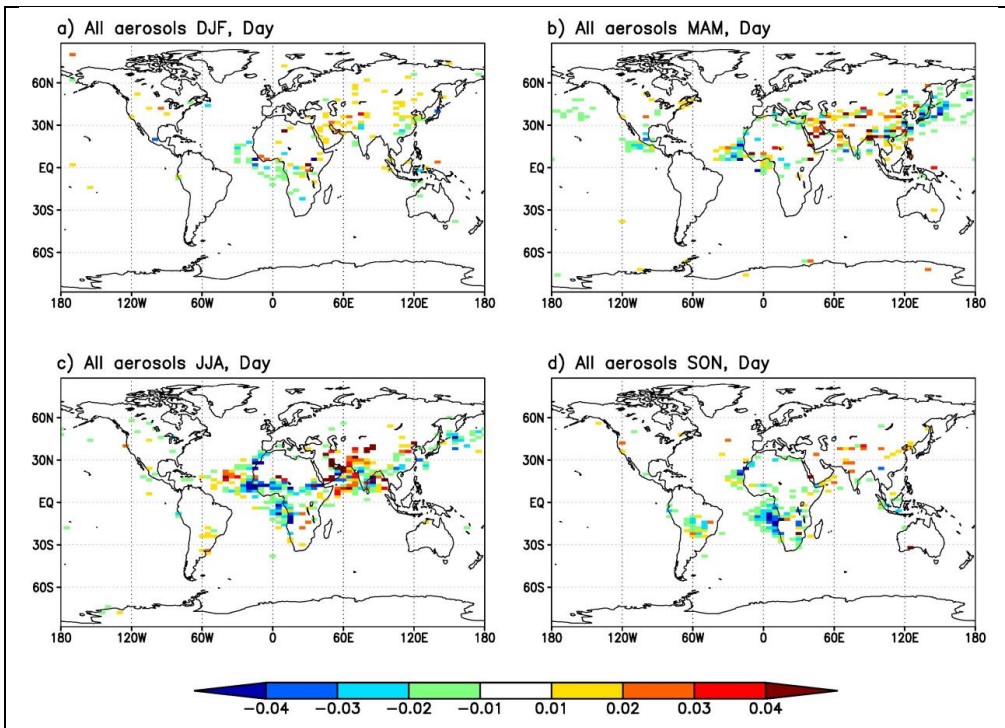

**Figure 2.** Daytime difference between AOD above the maximum low-cloud-top height over low clouds and the AOD above the same height in the clear sky neighborhoods for a) DJF b) MAM c) JJA and d) SON. Positive values mean that the aerosols above low clouds have a larger AOD than that in clear sky, and vice versa.





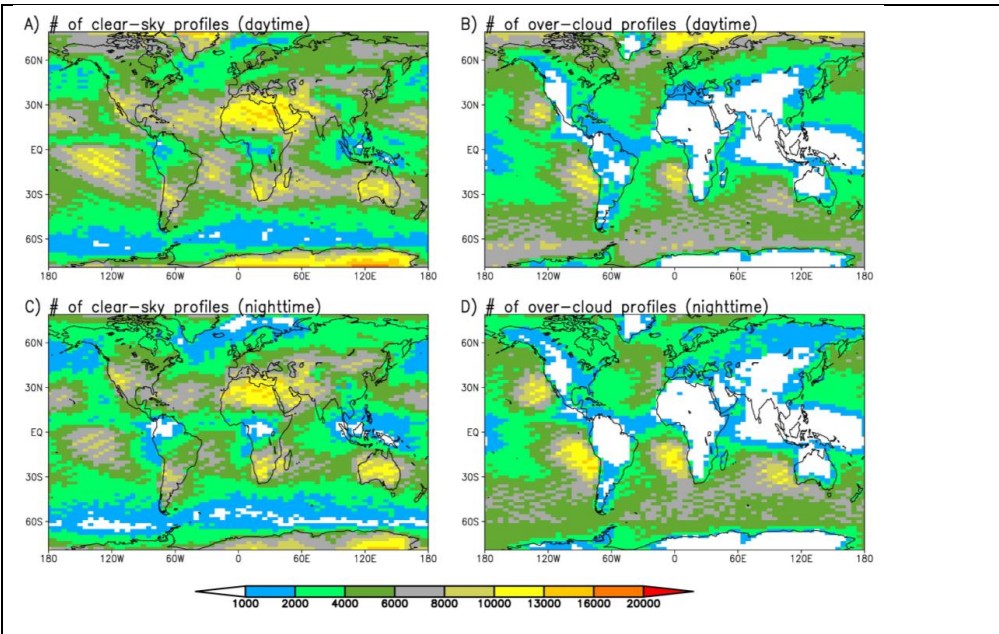

**Figure 3.** Number of profiles used for the analysis over each 2°×5° latitude-longitude gridbox. These numbers represent the total available profiles from June 2006 to October 2011.





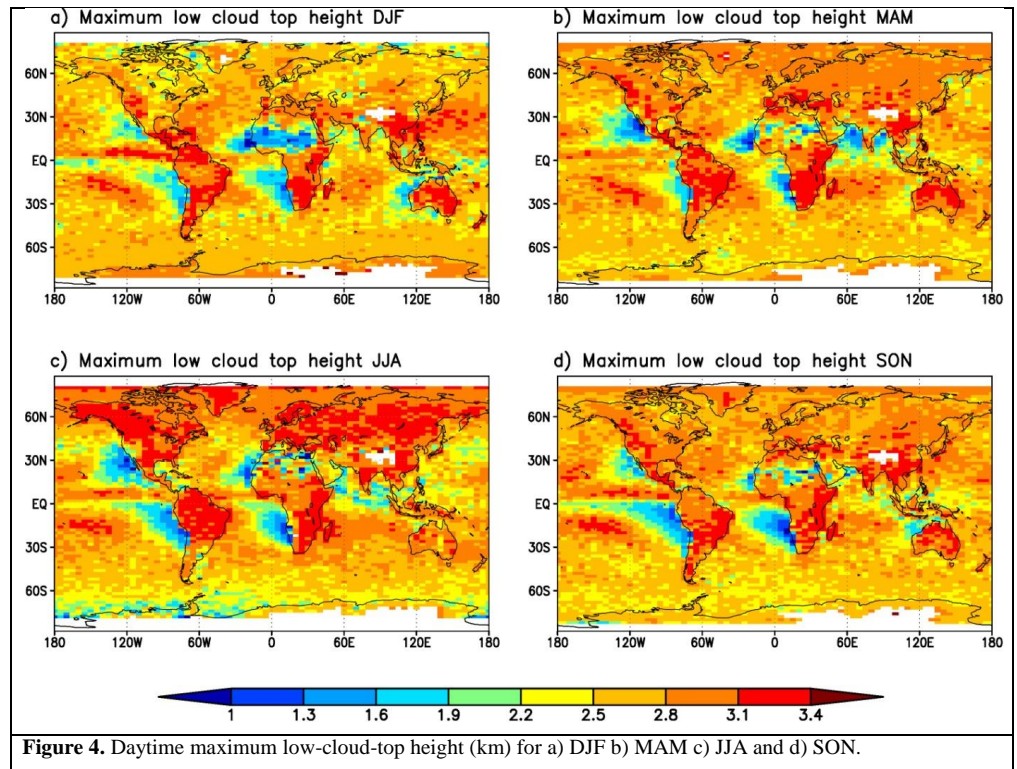

**Figure 4.** Daytime maximum low-cloud-top height (km) for a) DJF b) MAM c) JJA and d) SON.





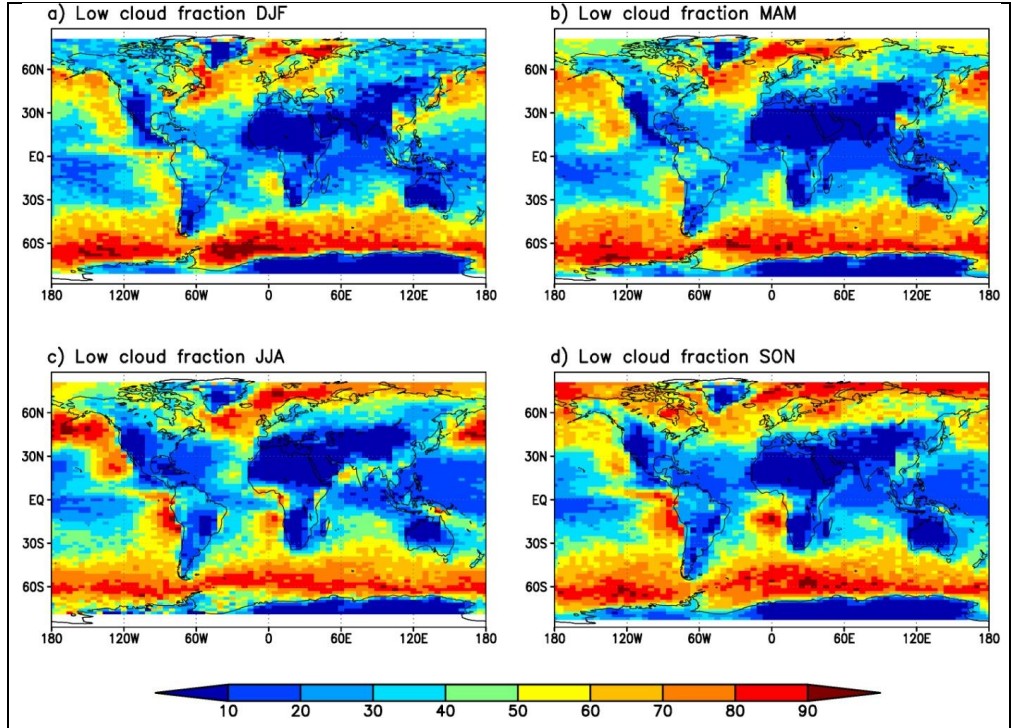

**Figure 5.** Daytime low cloud fraction (in %) calculated from profiles not containing middle and high clouds for
a) DJF b) MAM c) JJA and d) SON.





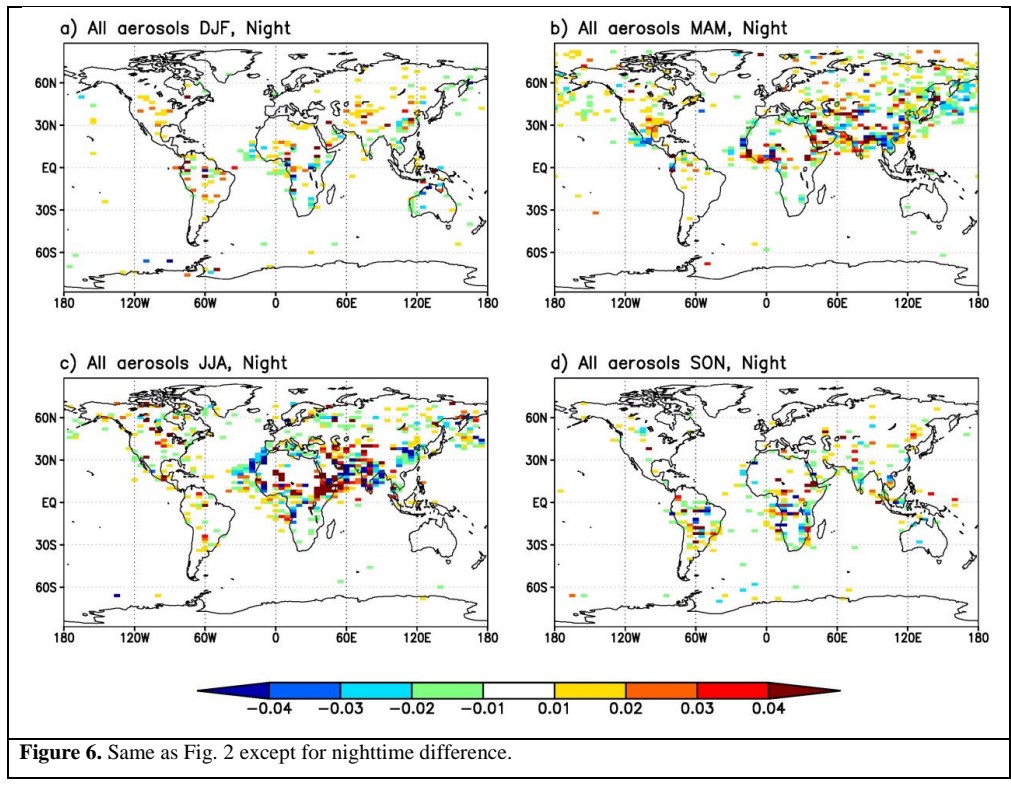

**Figure 6.** Same as Fig. 2 except for nighttime difference.





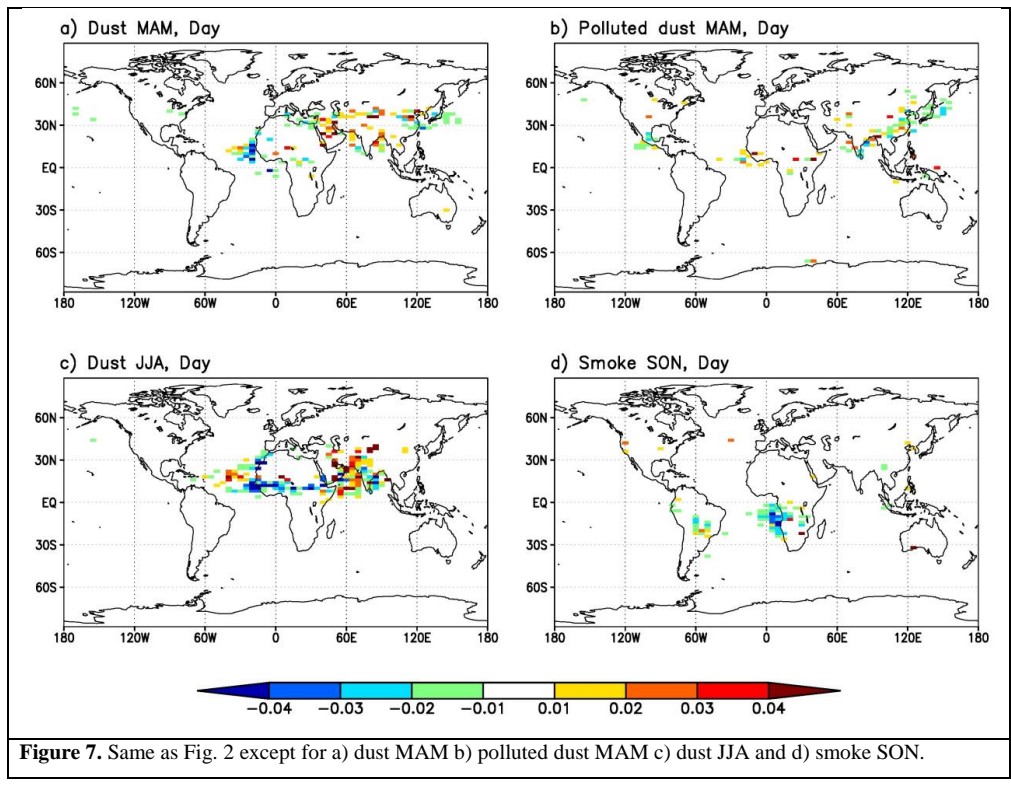

**Figure 7.** Same as Fig. 2 except for a) dust MAM b) polluted dust MAM c) dust JJA and d) smoke SON.