# Peer review of "Relationship between low-cloud presence and the amount of overlying aerosols"

_Atmospheric Chemistry and Physics, 2016_

## Referee Comment (RC1) · Anonymous Referee #1 · 26 Feb 2016

This paper study the difference between AODct in clear sky and above low clouds, presents some new and interesting results. In general, I find this manuscript to be of sufficient interest for publication and appropriate for ACP audience. I recommend accepted this paper for publication to do minor revisions that specified below:

1. The author should highlights why CALIPSO data are used in this study? Since there are some studies indicate that the CALIPSO derived above-cloud AOD are lower than other satellites sensors in the A-Train. 2. Do you consider the multilayer cloud with the existence of aerosols in this study? For example, aerosols above lower layer clouds but below the upper layer clouds? 3. When you calculate the ˑAODct, how do you define the nearby clear sky? As we know that for some grid cell, there may be three or two nearby clear sky grid cells. 4. Some discussions or conclusions should be given about the relationship between low-cloud presence and the amount

of overlying aerosols. 5. There are several related references advise you to cite in you study, for example: (1. Wang, W., J. Huang, P. Minnis, Y. Hu, J. Li, Z. Huang, J. Ayers, and T. Wang, Dusty cloud properties and radiative forcing over dust source and downwind regions derived from A-Train data during the Pacific Dust Experiment, Journal of Geophysical Research, 115 (2010), D00H35, doi:10.1029/2010JD014109. 2. Huang, J., P. Minnis, B. Lin, Y. Yi, S. Sun-Mack, T. Fan, and J. Ayers, 2006: Determination of ice water path in ice-over-water cloud systems using combined MODIS and AMSR-E measurements, Geophysical Research Letters, 33 (21)L21801, doi:10.1029/2006GL027038. 3. Huang, J., P. Minnis, B. Lin, Y. Yi, M. Khaiyer, R. Arduini, A. Fan, and G. Mace, 2005:Advanced retrievals of multilayered cloud properties using multispectral measurements, Journal of Geophysical Research, 110 (D15) (2005), D15S18, doi:10.1029/2004JD005101.) 6. In references, there are some unusual expressions. For example, some reference use "Atmos. Chem. Phys.", but some reference use "Atmospheric Chemistry and Physics".

---

## Author Comment (AC1) · 28 Feb 2016

We thank the reviewer for finding our paper interesting. We thank the reviewer for giving many valuable references. We will study these references and enhance the paper during the revision.

1. When we stated "the nearby clear sky", we were referring to the clear-sky portion in a gridcell. We were not referring to nearby clear sky gridcells. We will make this clearer during the revision.

2. The reviewer says "there are some studies indicate that the CALIPSO derived above-cloud AOD are lower than other satellites sensors in the A-Train". We are aware of potential CALIPSO data bias and will strengthen the discussion in this regard. However, to our knowledge, CALIPSO gives AOD for aerosols above cloud and no other

satellite gives such data. Other satellite sensors (such as MODIS) give clear-sky column average AOD. What satellites give the AOD for aerosols above cloud other than CALIPSO?

---

## Referee Comment (RC2) · Anonymous Referee #2 · 2 Mar 2016

This manuscript compares statistics of the CALIPSO derived aerosol optical depth (AOD) above low level clouds to that in cloud free scenes. Results are presented for both daytime and nighttime retrievals of AOD. The work illustrates that there are some differences in the AOD above low cloud and the AOD in clear skies in the daytime retrievals e.g. over the southeast Atlantic in SON. These are however not apparent in the nighttime retrievals. This is very suggestive of a bias in the satellite retrieval of AOD during the daytime.

The authors clearly have some ideas as to why the retrieval scheme could have issues during the daytime (detection threshold in the feature detection algorithm, underlying surface albedo effects etc). The paper would be strengthened significantly if these were explored further and perhaps investigated on a case-study basis e.g. a CALIPSO track in the southeast Atlantic with significant biomass burning AOD above low level

cloud and in cloud free scenes.

My judgment is that for the paper to be suitable for publication in ACP, additional work that examines the retrieval scheme in more detail to better understand where the day-time differences originate from are required.

---

## Author Comment (AC2) · 2 Mar 2016

We thank the reviewer for well understanding the key point of the paper. About the AOD difference over the southeast Atlantic, as the reviewer suspected, this could be due to a bias in retrieval algorithm. This could also be a real phenomenon as the cloud over this part of the world has a strong diurnal cycle. Examining this aspect to the full (especially the retrieval algorithm) would require a multi-year study and another paper or two in our judgement. Thus, for our paper, exposing such problems suffices.

---

## Author Response (AR1)

This paper study the difference between AODct in clear sky and above low clouds, presents some new and interesting results. In general, I find this manuscript to be of sufficient interest for publication and appropriate for ACP audience. I recommend accepted this paper for publication to do minor revisions that specified below:
➔ We thank the reviewer for finding the paper acceptable for ACP.

1. The author should highlights why CALIPSO data are used in this study? Since there are some studies indicate that the CALIPSO derived above-cloud AOD are lower than other satellites sensors in the A-Train.
➔ During the revision, we added the following texts: "We calculate 532 nm AOD above cloud-top-height and this AOD differs from the AOD above cloud retrieved by the CALIOP operational algorithm. Furthermore, we compare the AOD above low cloud top height with that at the same height in the nearby clear sky, and this comparison is not possible with data retrieved passive sensors even though passive sensors could provide the AOD above cloud. Thus, the use of CALIPSO retrievals is necessary for our investigation." (section 1)

2. Do you consider the multilayer cloud with the existence of aerosols in this study? For example, aerosols above lower layer clouds but below the upper layer clouds?
➔ Addressing aerosols above low cloud but below high or middle cloud is entirely a new issue, and CALIPSO operational algorithm retrievals are not sufficient. In other words, the request is beyond the scope of our present study. We added the following text: "Thus, we are not addressing the aerosols above low cloud but below high or middle cloud. Detecting multi-layer clouds from satellite sensors is possible (e.g., Huang et al., 2005; Huang et al., 2006) but addressing aerosols below cloud in CALIPSO retrievals creates additional uncertainties and challenges that are not very necessary to investigate the relationship between aerosol amount and underlying cloud." (section 3)

3. When you calculate the ∆AODct, how do you define the nearby clear sky? As we know that for some grid cell, there may be three or two nearby clear sky grid cells.
➔ In the paper, we were referring to the clear-sky portions within a grid cell, relative to the cloudy-portions within the same grid cell. To make this clearer, we revised the paper in the following:
At Fig. 2 caption, "Positive values mean that the aerosols above low clouds have a larger AOD than that in the clear sky portion of the grid cell."
In section 1, "This procedure thereby assumes that the AOD is independent of clouds and that the AOD above clouds is the same as that at the same height in the clear sky neighborhoods within the grid cell."
In section 5, "That said, we have offered the observational quantification of the difference between AOD above cloud-top height in the clear-sky portions of a 2°×5° gridbox and in the low-cloud occupied portions of the same gridbox over the globe."

4. Some discussions or conclusions should be given about the relationship between low-cloud presence and the amount of overlying aerosols.
➔ We added a para at the end of section 5.

5. There are several related references advise you to cite in you study, for example:
➔ We added the following texts in section 3: "Thus, we are not addressing the aerosols above low cloud but below high or middle cloud. Detecting multi-layer clouds from satellite sensors is possible (e.g., Huang et al., 2005; Huang et al., 2006) but addressing aerosols below cloud in CALIPSO retrievals creates additional uncertainties and challenges that are not very necessary to investigate the relationship between aerosol amount and underlying cloud."

**Anonymous Referee #2**

This manuscript compares statistics of the CALIPSO derived aerosol optical depth (AOD) above low level clouds to that in cloud free scenes. Results are presented for both daytime and nighttime retrievals of AOD. The work illustrates that there are some differences in the AOD above low cloud and the AOD in clear skies in the daytime retrievals e.g. over the southeast Atlantic in SON. These are however not apparent in the nighttime retrievals. This is very suggestive of a bias in the satellite retrieval of AOD during the daytime.

The authors clearly have some ideas as to why the retrieval scheme could have issues during the daytime (detection threshold in the feature detection algorithm, underlying surface albedo effects etc). The paper would be strengthened significantly if these were explored further and perhaps investigated on a case-study basis e.g. a CALIPSO track in the southeast Atlantic with significant biomass burning AOD above low level cloud and in cloud free scenes.

My judgment is that for the paper to be suitable for publication in ACP, additional work that examines the retrieval scheme in more detail to better understand where the daytime differences originate from are required.

➔ We thank the reviewer for well understanding the key point of the paper. About the AOD difference over the southeast Atlantic, as the reviewer suspected, this could be due to a bias in retrieval algorithm. We acknowledged this possibility in the manuscript. This could as well be a real phenomenon as the boundary layer cloud over this part of the world has a strong diurnal cycle. Examining this aspect to the full (especially changing retrieval algorithms) would require a multi-year study and another paper or two in our judgement. Plus, making an unbiased comparison of daytime $AODct$ above low clouds and in clear sky may not solve the problem if this reduces the $\Delta AODct$ day and night discrepancy over the southeast Atlantic but preserves the $\Delta AODct$ day and night discrepancy and makes this discrepancy statistically insignificant. In view of this, for our current paper, we contend that exposing the $\Delta AODct$ day and night discrepancy over this region (and other regions) suffices.

During the revision, we strengthened the paper with regard to the $\Delta AODct$ day and night discrepancy. See "Another candidate is aerosol-cloud interaction. Aerosols were shown to influence underlying cloud by indirect effects and semi-direct effects. For example, by analysing satellite observations, Costantino and Bréon (2010; 2013) found that cloud droplet size decreases with aerosol loading when the bottom of an aerosol layer touches the top of the cloud beneath. In contrast, when the aerosol layer is well separated from the underlying cloud, they found no correlation between cloud droplet size and aerosol loading. This indicates that some of the

aerosols entrained from cloud top when the aerosol layer touches cloud top becomes activated as CCN and increases the cloud droplet number concentration and decreased the cloud droplet size. In addition to this indirect effect mechanism, overlying aerosols, in case of sunlight absorbing aerosols, can create semi-direct effects by the short-wave radiative aerosol heating. This heating can increase the temperature inversion above the boundary layer, which helps to inhabit the entrainment of free-tropospheric dry air through cloud top and thereby produces a moister boundary layer and thus thicker, more reflective clouds (Johnson et al., 2004; Wilcox, 2010). This temperature inversion is enhanced by the fact that atmospheric solar heating by an aerosol layer increases when the layer resides above cloud. These aforementioned aerosol-cloud interactions and possibly more are expected to somehow affect the aerosol amount over cloud by, e.g., influencing aerosol deposition." (section 1)

Also, see "On the other hand, over the northwest Pacific Ocean in MAM, the nighttime results are in agreement with the daytime results that $AODct$ above low clouds is lower than that in clear sky. It is very difficult to imagine that the possible bias in CALIPSO retrieval is limited to the southeastern Atlantic Ocean. In fact, boundary layer clouds in the southeastern Atlantic Ocean are known to exhibit a strong diurnal cycle (Rozendaal et al., 1995; Wood et al., 2002; Min and Zhang, 2014). Thus, the $\Delta AODct$ day and night discrepancy over this region could be a real phenomenon rather than a byproduct of imperfect CALIPSO retrievals, due to the aerosol-cloud interaction through short-wave radiation, as discussed in section 1. There is no short-wave radiation in the night." (section 5)